# Application of a Magnetic Field to Enhance the Environmental Sustainability and Efficiency of Microbial and Plant Biotechnological Processes

**Miroslava Sincak [1], Alena Luptakova [2], Ildiko Matusikova [1], Petr Jandacka [3] and Jana Sedlakova-Kadukova [1,\*]**

[1] Faculty of Natural Science, University of Ss. Cyril and Methodius in Trnava, Nam. J. Herdu 2, 91701 Trnava, Slovakia; sincak1@gmail.com (M.S.); ildiko.matusikova@ucm.sk (I.M.)

[2] Institute of Geotechnics, Slovak Academy of Sciences, Watsonova 45, 04001 Kosice, Slovakia; luptakal@saske.sk

[3] Faculty of Forestry and Wood Sciences, Czech University of Life Sciences Prague, Kamycka 129, 16500 Praha, Czech Republic; petr.jandacka@gmail.com

\* Correspondence: prof.jana.sedlakova@ucm.sk

**Abstract:** Despite the growing prevalence of using living organisms in industry, the control of biotechnological processes remains highly complex and constitutes one of the foremost challenges in these applications. The usage of electromagnetic fields offers a great opportunity to control various biotechnological processes by alternating growth and cell metabolism without influencing the characteristics of the cultivation medium or the products of the biotechnological process. The investigation of electromagnetic field applications across various industries, including food production, medicine, and pollutant mitigation, has yielded substantial insights. We used the scientific databases PubMed and ScienceDirect to select 103 experimental and theoretical articles that included original results suitable for further investigation. This type of search was repeated with every new relevant article iteratively until no new articles could be detected. Notably, even weak, low-frequency magnetic fields can accelerate the growth of certain organisms, further stabilize the bacterial community in activated sludge within wastewater treatment plants, enhance the fermentation capabilities of both yeast and bacteria, enhance metal bioleaching by the activation of bacterial metabolism, or improve the metal tolerance of plants during the phytoremediation process. Moreover, magnetic fields exhibit a promising sustainable possibility for the better control of biotechnological processes, thus making these processes more competitive compared with the currently used long-term unsustainable extraction of metals. Although with these interesting results, these examples represent highly exceptional applications. Despite these examples, the overall application potential of magnetic fields remains largely unexplored and unknown.

**Keywords:** electromagnetic field; biotechnology; microorganisms; metal

## 1. Introduction

During the process of evolution, the Earth's magnetic field constituted a natural component of organisms' environments. The Earth's magnetic field is approximately 25–65 µT at the equator and can exceed 60 µT near the magnetic poles, resulting from variations in magnetic field lines' inclination [1]. The outer shield of the Earth's magnetic field plays a critical role in protecting our planet from harmful solar and cosmic rays. It acts as a protective barrier that deflects and traps high-energy charged particles, preventing them from directly impacting the Earth's surface. This shielding effect is essential for maintaining a habitable environment, as exposure to these particles can be harmful to living organisms [1,2].

However, in the present era, numerous organisms have had to adapt to different magnetic fields generated mainly by electrical devices. The ubiquitous nature of electromagnetic radiation and its as yet unknown effects on humans raise concerns about public

health. Apart from the anticipated adverse effects, magnetic fields also possess several beneficial properties. In contemporary human medicine, they are used for cancer treatment [2]. Furthermore, they have potential applications in biotechnological processes, where the usage of living organisms represents a challenge in process control.

Maintaining optimal conditions for growth is essential for organisms, and prioritizing cellular processes that lead to better product yields (e.g., drugs, enzymes, and other biomolecules) or the acceleration of desired processes is crucial for industries [3]. In this context, the magnetic field can serve as a valuable tool in biotechnological process control, acting as an external environmental factor across a wide industrial spectrum due to its non-invasive nature. The usage of magnetic fields can also present a long-term sustainable alternative that is environmentally acceptable (while using magnets or weak electric fields) and has the potential to influence a broad spectrum of biotechnological processes, thereby enhancing their competitiveness compared to traditional less ecological methods [2,3].

In the past, a few authors have proposed that electromagnetic fields could be another tool that is capable of influencing the biotechnological process in a desirable way [1]. The usage of magnetic fields can have broad applications across various biotechnologies, as it does not directly affect the characteristics of the medium and, therefore, does not influence the composition of the medium or the quality of the product. However, it can accelerate growth and biomass production, stabilize bacterial communities, and expedite certain processes such as waste processing or metal recovery [4]. Despite the explored potential of magnetic fields to influence certain cellular processes, there are still limited studies that have attempted to use this potential in practice, and this area deserves further investigation in the future. Implementing an appropriate control mechanism is one promising approach to make biotechnological processes competitive alternatives to traditional industrial procedures [5]. A summary of the existing results on the usage of magnetic fields for controlling biotechnological processes is provided in the following text.

The aim of this article is to summarize the already-published information about the usage of electromagnetic fields in the control of biotechnological process with a focus on plant and microbial technologies. In this review, we discuss not only the importance and challenges of biotechnological process control, but also summarize the information published about the potential of magnetic fields to be used in eukaryotic biotechnology, specifically medicine, the food industry, as well as phytoremediation. We also gather information about prokaryotic biotechnology, including metal bioleaching, medicine, the food industry, and pollutant removal and discuss how they can be improved by magnetic field exposure.

## 2. Methods

We used the scientific databases PubMed and ScienceDirect to select papers containing the keywords 'magnetic' and 'electromagnetic,' as well as 'industry,' 'medicine,' 'biotechnology,' or 'environmental biotechnology' in their titles, abstracts, and keywords. After a subsequent semantic verification process, we identified 96 experimental and theoretical articles featuring original findings suitable for further investigation. These articles spanned the years 1997 to 2023, and we had a particular focus on newly published works. This search process was iteratively repeated to incorporate any newly relevant articles until no further ones could be identified.

## 3. Control of Biotechnological Process

Nowadays, the usage of living organisms in industrial applications is a welcomed environmentally acceptable alternative to some industrial methods and chemical synthesis processes. The importance of bacterial technologies in the pursuit of sustainable industrial processes is undeniable [5,6]. Incorporating living organisms into industrial processes not only offers environmentally friendly alternatives and the potential for long-term sustainability, but also addresses the persistent challenge faced by today's industries of controlling biotechnological processes. Despite the undeniable advantages of biotechnologies, such as

their environmentally friendly approach, low cost, substrate flexibility, and low or lack of waste production, etc., the control of biotechnological processes remains challenging for industries today [4]. The main reasons for this are illustrated in Figure 1.

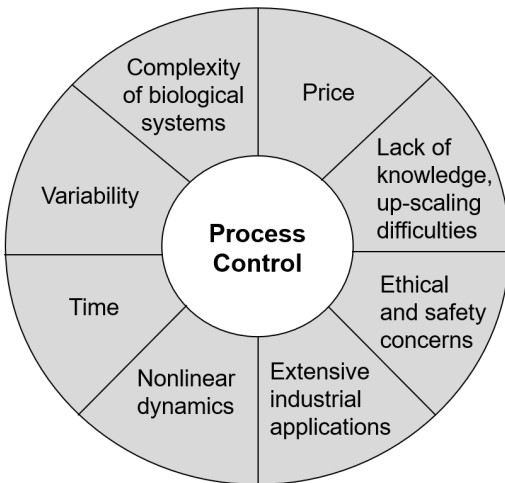

**Figure 1.** The challenges of biotechnology process control.

- Complexity of biological systems: Organisms commonly used in biotechnological applications are complex biological systems; therefore, it is difficult to predict how specific changes in the environment will affect the organism at multiple levels, and even subtle changes can lead to inconsistent outcomes or alterations in the quality or quantity of the resulting product [6].
- Variability: Biological systems used in biotechnology include bacteria, fungi, plants, as well as higher organisms. Therefore, a uniform method of controlling processes is impossible, and each biotechnological application, along with each organism used within it, requires specific conditions for the optimal growth and production of the target product [7,8].
- Nonlinear dynamics: Biological systems often exhibit nonlinear dynamics, meaning that small changes in one part of the system can have disproportionately large effects on the overall behavior of the system. This effect can make it difficult to predict how a biotechnological process will respond to changes in external or internal parameters [9,10].
- Lack of knowledge and up-scaling difficulties: Despite advances in biology, much remains unknown about how biological systems function and precisely respond to changes in the environmental conditions at the physiological or genetic levels. This can pose challenges in designing and controlling biotechnological processes with a high degree of precision [3]. It is important to ensure that the process can operate at larger volumes without losses in the efficiency or product quality [11].
- The price: Biotechnological processes used in certain industries, such as the pharmaceutical industry, may be less cost-effective compared to traditional industrial processes [12]. Furthermore, these processes often require specialized equipment to provide the optimal conditions for the used organisms, which can further increase their costs [13]. Seeking cost-effective strategies for the control of biotechnological processes is one important approach for making biotechnological processes more accessible and competitive [14].
- Time: Certain biotechnological processes, such as protein or metabolite production using microorganisms, bacterial bioleaching, and microbial waste treatment, can, in some cases, be slower compared to conventional processes [5]. This can pose a challenge for the industrial implementation of these biotechnological processes, which often require large quantities of a product in a short period [4].

- Ethical and safety concerns: Ethical and safety considerations still exist and must be taken into account when working with biological systems. The potential risks and unintended consequences of biotechnological processes can be subjects of public concern, which can present challenges in balancing the benefits of a particular process with its potential risks [15].
- Extensive industrial application of biotechnology: The use of living systems is not limited to a single industry but can be found in nearly every sector, from healthcare to environmental remediation and waste processing [16]. Since each of these industries uses different organisms and produces different target products, they require a varying process control complexity and face different challenges [17].

Due to the complexity of managing biotechnological processes, we currently have only a few main options for controlling them. These options include the use of various substrates for microbial growth, the monitoring of external conditions and process automatization to increase the efficiency, the employment of specific microorganism strains, or the use of genetic engineering to enhance the ability of microorganisms to perform specific tasks.

## 4. The Application of Magnetic Fields in Biotechnological Processes Using Eukaryotic Organisms

Notably, research has revealed that magnetic fields possess the potential to expedite specific processes, as exemplified by enhanced ethanol production and plant-based phytoremediation. Beyond these applications, the influence of magnetic fields on human immunity and their potential employment in medical treatments have also emerged. These sectors are intricately detailed in the subsequent subsections to allow a comprehensive understanding. The diverse applications of magnetic fields in these areas are summarized in Table 1.

**Table 1.** The effects of magnetic fields on biotechnological processes using eukaryotic organisms.

| Impact on | Effect | Organism | Magnetic Field Properties | | | | |
|---|---|---|---|---|---|---|---|
| | | | Magnetic/ Electromagnetic Field | B (mT) | Frequency (Hz) | Duration | References |
| cancer treatment | promotion | mice | alternating | not stated | 230 kHz | 2–8 days | [18] |
| | promotion | cancer cell cultures | alternating | 5.1 | 50 Hz | 2 h in 3 days | [19] |
| bone healing | promotion | dogs | static | 100 | - | 8 weeks | [20] |
| wound healing | promotion | mice | static | 5 | 25 Hz | 1h/10 days | [21] |
| M1 macrophages transformation to macrophages M2 | promotion | mouse macrophages | static | 1240 | - | not stated | [22] |
| immune response | promotion | human tumor cells | alternating (rotatory) | not stated | 2 Hz | 10 min | [23] |
| metal accumulation | promotion | *Noccaea caerulescens* | alternating | 30, 60, 120, 150 | not stated | 20 min/7 days | [24] |
| | promotion | *Noccaea caerulescens* | alternating | 400 | not stated | 20 min/7 days | [24] |
| phytoremediation | promotion | *Eucaliptus globulus* | static | 30, 60, 120, 150 | - | 20 min/7 days | [25] |
| | promotion | *Celosia argentea* | static | 30–150 | - | 20 min/7 days | [26] |
| | promotion | *Eucaliptus globulus* | static | 400 | - | 20 min/7 days | [25] |
| | no effect | *Celosia argentea* | static | 30–150 | - | 20 min/7 days | [26] |
| | promotion | water for watering of *Festuca arundinacea* | static | 100 | - | 10 s/5 min | [27] |
| wastewater treatment | promotion | *Scenedesmus obliquus* | static | 50–500 | - | 3 days | [28] |
| ethanol production | promotion | *Saccharomyces cerevisiae* | alternating | 8–10 | <50 Hz | 24 h | [29] |
| | promotion | *Saccharomyces cerevisiae* | static | 200 | - | 24 h | [30] |
| | no effect | *Saccharomyces cerevisiae* | alternating (rotatory) | 1 | 100 Hz | not stated | [31] |
| fermentation | promotion | *Geotrichum candidum* | static | 7 | - | 24–120h | [32] |
| mycotoxins content | no effect | *Saccharomyces cerevisiae* | static | 35 | - | 48h | [33] |

### 4.1. The Food Industry and Ethanol Production

*Saccharomyces cerevisiae* is the most well-known microorganism involved in ethanol production and has been used in fermented beverages for centuries [34,35]. Under anaerobic conditions, *S. cerevisiae* utilizes pyruvic acid derived from sugar catabolism, which is converted via acetaldehyde into ethanol. A total of 12 enzymes are involved in the fermentation process. However, the two key enzymes in the yeast fermentation pathway are pyruvate decarboxylase and alcohol dehydrogenase, which regenerate $NAD^+$ and produce ATP [35]. The choice between the aerobic and anaerobic glucose degradation pathways is genetically encoded and relies on gene repression induced by the Crabtree effect, which modifies the respiratory chain of the Krebs cycle and switches glucose metabolism to an anaerobic process that occurs in the cytosol instead of aerobic metabolism in the mitochondria [30]. The alcohol fermentation process spontaneously ends when the ethanol concentration in the medium reaches approximately 14 to 20%. Such high levels of ethanol have an inhibitory effect on the yeast cells that produce it. The inhibitory effect of ethanol lies in changes in the organization and permeability of the cell membrane and the deactivation of cytosolic enzymes [31]. During industrial alcohol fermentation, it is crucial to maintain yeast cells in a reproductive (budding) state and to avoid stressful conditions that could jeopardize their growth [36].

In the study of the influence of a magnetic field on the fermentation process, Perez et al. [29] observed the positive effect of extremely low-frequency magnetic fields with a magnetic induction of B = 5–20 mT on ethanol production. The conditions with a magnetic field of 20 mT showed the best results, with the total ethanol productivity being approximately 17% higher than in the control experiment. There was also a difference in the fermentation time, as the yeast exposed to the magnetic field reached its final stage approximately 2 h earlier than the control experiment.

The effect of a magnetic field (permanent magnet, 220 mT) was also studied by Da Motta et al. [30]. The authors documented a 3.4-fold increase in the ethanol concentration compared to the control culture. Glucose consumption was also higher, correlating with the ethanol yield. Even after 24 h, the yeast exposed to the magnetic field continued to proliferate intensively, despite the high alcohol content, unlike the control culture. Alcohol production in magnetized cultures steadily increased from the fourth to the twenty-fourth hour, resulting in a production rate 1.68 times higher than in cells not exposed to a magnetic field.

Anaerobic alcohol fermentation is known to be directly related to the production of gaseous $CO_2$ and increased biomass production [37]. Therefore, the authors hypothesized that the mechanisms of ethanol-to-glucose conversion are likely influenced by the static magnetic field, as the glucose/ethanol/biomass conversion rate in cultures exposed to the magnetic field was higher than in the control.

In contrast to the positive results with static and low-frequency magnetic fields [29,30], the rotating magnetic field alone was not able to increase ethanol production [31]. An increase in the fermentation rate was observed only in combination with a magnetic field (amplitude: 1 mT, f = 100 Hz) and magnetic particles in the medium, resulting in a 50% increase [31]. These different observations confirm the assumption that the type and strength of the magnetic field significantly influence the resulting biological effect on living organisms [38].

In addition to its impact on the alcohol product itself, there is a possibility of using magnetic fields to regulate the content of mycotoxins present in plant-based fermentation substrates using *S. cerevisiae* yeast, which has the ability to degrade certain contaminants during fermentation through the activity of specific yeast enzymes [39]. These enzymes are involved in cellular physiological processes related to protection against oxidative stress and detoxification [40], and their activity can be enhanced by the application of a magnetic field [41]. Therefore, alcohol fermentation with *S. cerevisiae* holds promise as a method for reducing the mycotoxin content [39]. However, Boeira et al. [33] did not observe a significant change in the mycotoxin concentration after the application of a magnetic field

(35 mT) to the fermentation process. The potential of magnetic fields to aid in the removal of fermentation contaminants has been the subject of only a small number of studies, and more attention should be given to exploring this issue in the future.

Improving the fermentation capabilities of selected fungi is another area of research where magnetic fields have the potential to be used to enhance biomass production and the production of specific enzymes. Canli et al. [32] investigated strains of *Geotrichum candidum* and its fermentation capabilities. In their experiment, they significantly increased the enzyme activity of inulinase and biomass production through the application of a static magnetic field of 7 mT. Based on these results, the authors considered the magnetic field to be an effective method for increasing the production of specific products by various fungi.

### 4.2. Medical and Laboratory Applications

Magnetic fields can be also used as an adjunctive therapy for the treatment of certain diseases, but the most studied remains the auxiliary medical procedure for cancer treatment. Recently, the use of circularly polarized magnetic fields has been investigated to enhance the immune response by increasing tumor cell death and accelerating the maturation of dendritic cells and the infiltration of T-lymphocytes in the tumor [42]. Furthermore, the combination of magnetic-field-induced hyperthermia and the integration of nanoparticles containing anticancer drugs has been studied. Clusters of $Fe_3O_4$ nanoparticles generated heat upon electromagnetic field application, leading to the release of doxorubicin. According to the authors, this combination has the ability to destroy cancer cells and achieve a complete cure without malignancy recurrence [23]. Increased levels of cell death have also been observed by other researchers studying electromagnetic fields (50 Hz) modulated by a static magnetic field (5.1 mT). The more frequent occurrence of apoptosis and ferroptosis is attributed to the induction of reactive oxygen species (ROS) and significant DNA damage, as well as the activation of DNA repair pathways. This combination of magnetic field effects had an inhibitory impact on the population of cancer cells [19].

Magnetic fields have also been studied in relation to patient immunity. The exposure of M1 pro-inflammatory macrophages to non-uniform magnetic fields causes extreme elongation of macrophages and the acquisition of an anti-inflammatory M2 macrophage phenotype. This transformation depends on the position relative to the magnetic field lines [22].

Other applications of magnetic fields can be to accelerate wound healing [20,43] or as an adjunctive treatment for arthritis [44], as well as to initiate the differentiation and migration of stem cells [45]. In the past, the abilities of electric, magnetic, and electromagnetic fields to aid in wound healing and associated inflammatory processes have been studied [46]. Some studies also highlight the effects of electromagnetic fields and their use in tissue regeneration. It has been found that frequencies and intensities of pulsed electromagnetic fields in the range of <100 Hz and 3 mT have positive effects on accelerating wound healing processes [47]. Clinical studies in humans have also demonstrated that electromagnetic fields reduce the healing time and recurrence rate of leg ulcers [48] and may have anti-inflammatory effects [21].

In addition to the direct use of magnetic fields in medical therapy, the application of orthogonal electric pulses with durations of 0.1–2 ms and field intensities of 2.5–4.5 kV/cm to a yeast suspension led to the release of cytoplasmic proteins without cell lysis, which aided in protein extraction. Treated cells were more susceptible to enzymatic cleavage. Depending on the strain and electric conditions, cell lysis was achieved at a 2- to 8-fold lower enzyme concentration compared to the control. These findings could be useful for the efficient isolation of proteins from cells without complete cell lysis [49].

### 4.3. Pollutant Removal

Soil remediation is another potential but underexplored area where magnetic fields could be applied as a low-cost biotechnological enhancement. Compared to the fields

used in yeast and prokaryotic organisms, the fields investigated in plants are much higher (100–400 mT), and lower fields have been deemed to be insufficiently effective.

Several studies have observed the beneficial effects of magnetic fields on the phytoremediation process of metal-contaminated soil. The results show that the application of magnetic fields improves the soil remediation efficacy with a significantly reduced total Cd content (38.9%) and bioavailable Cd content (27.3%) in the soil. Additionally, the Cd content in two types of rice grains was significantly reduced [50]. Further research has demonstrated that plants grown from magnetic field pre-treated seeds have the ability to accumulate 28.8–250.1% more metals (Cu, Cd, Hg, Pb, Zn, Cr) [24,25]. The authors consider the best conditions for enhancing phytoremediation to be 120 mT for *Noccaea caerulescens* and 150 mT for *Eucalyptus globulus* [25] and 100 mT for *Celosia argentea* [26], which exhibited the greatest increases in biomass and accumulated metals.

For *E. globulus* and *N. caerulescens*, a magnetic field strength of 400 mT was found to be inhibitory in terms of biomass production as well as for the accumulation of present metals [24,25]. According to Yang et al. [26], a magnetic field strength of 30 mT is insufficient to affect the phytoremediation process using *Celosia argentea*.

In terms of economics, the costs associated with phytoremediation are related to the need for irrigation of the used plants. Prolonged drought can affect the transpiration rate and the efficiency of Cd extraction from the soil, inducing oxidative damage in plant cells. Yang et al. [26] reported that pre-treating seeds with a magnetic field helped plants to overcome 3- to 10-day drought periods and had a positive impact on the phytoremediation process, biomass production, and pigment levels, thus alleviating the harmful effects caused by drought.

Another economic challenge is the harvesting of whole plants after the completion of remediation. Harvesting above-ground parts such as leaves and stems is considerably easier. Leaves, in particular, are the site of intensive cadmium accumulation in *Festuca arundinacea*. According to Luo et al. [27], irrigating the plants of *Festuca arundinacea* with magnetized water increased the biomass of aging and dead leaves, which then redistributed significantly higher amounts of Cd compared to a control (approximately 23.6% higher compared to the control). Based on these findings, it can be stated that treating seeds or whole plants with a magnetic field has the potential to become a novel economic strategy to enhance the efficiency of phytoremediation.

In addition to the direct influence on plants, an innovative approach using the symbiotic relationship between bacteria from wastewater and the microalga *Scenedesmus obliquus* has been proposed to improve wastewater treatment, along with the application of a static magnetic field [51]. The algae–bacterial symbiotic system can enhance the production of dissolved oxygen, thereby promoting bacterial growth and the catabolism of pollutants in wastewater. The results showed that the magnetic field (ranging from 50 to 500 mT) stimulated algal growth, increased oxygen production by 24.6%, and elevated the chlorophyll content by 11.5% compared to the control. The study confirmed that the application of an appropriate magnetic field could reduce the energy consumption required for aeration in the degradation of organic matter in municipal wastewater using the symbiotic system.

## 5. Application of Magnetic Fields in Biotechnological Processes Involving Prokaryotic Organisms

Compared to eukaryotic organisms, more studies have been dedicated to prokaryotic organisms, likely due to their simpler cellular structure, rapid response to environmental changes [52], and wide industrial applications. Preliminary findings regarding the use of electromagnetic fields for controlling prokaryotic organisms encompass various areas, including the food industry, wastewater treatment plants, biofilm removal, the enhancement of antibiotic sensitivity to antibiotics, as well as bacterial metal recovery. These sectors will be further discussed in the following subsections and are summarized in Table 2.

**Table 2.** The effects of magnetic fields on biotechnological processes using prokaryotic organisms.

| | | | Magnetic Field Properties | | | | |
|---|---|---|---|---|---|---|---|
| Impact on | Effect | Organism | Magnetic/ Electromagnetic Field | B (mT) | Frequency (Hz) | Duration | References |
| nitrogen removal, bacterial survival | promotion | active sludge | static | 88 | - | 12 h | [53] |
| | reduction | active sludge | static | 88 | - | 6 days | [53] |
| denitrification | promotion | active sludge | static | 39.5–65.3 | - | 8 and 12 h | [54] |
| | promotion | active sludge | static | 30 | - | 8 and 12 h | [55] |
| concentration of COD * | promotion | active sludge | static | 350 | - | 48 h | [56] |
| nitrogen removal | no effect | active sludge | static | 350 | - | 48 h | [56] |
| contaminants removal ** | promotion | *Pseudomonas* spp., *Cupriavidus* spp., *Rhodococcus* spp. | static | 200 | - | 5 h | [57] |
| lignocellulose degradation | promotion | environmental microorganism | alternating | 230~260 | electric field (0.3~0.8 V) | 3–21 days | [58] |
| oil removal | promotion | *Acinetobacter* sp. B11 | static | 25 | - | 7 days | [59] |
| vinegar aging | promotion | vinegar-producing bacteria | alternating | 0–5 | not stated | 3 h | [60] |
| biofilm removal | promotion | *Pseudomonas aeruginosa* | static and alternating | 444 | 474 kHz | 3–24 h | [61] |
| biofilm removal | promotion | *Enterococcus faecalis* | static | 170 | - | 24–72 h | [62] |
| | reduction | bacteria of teeth biofilm | static | 60 | - | 24 and 48 h | [63] |
| hydrogen production | promotion | *Clostridium pasteurianum* | static | 3.2 | - | 60 h | [64] |
| bioleaching (Cu, Fe) | promotion | *Acidithiobacillus ferrooxidans* | static | 3.14 | - | 24 days | [65] |
| bioleaching (As, Cd) As a Cd | promotion | *Acidithiobacillus ferrooxidans* | static | 8–10 | - | 30 min | [66] |
| bioleaching (Cr) | promotion | *Geotrichum* sp. a *Bacillus* sp. | static | 7 | - | 24 h | [67] |
| bioleaching (Cu) | promotion | *Acidithiobacillus ferrooxidans, Thiobacillus thiooxidans* | static | 9.6 | - | 27 days | [68] |
| bioleachin (Cu) | promotion | mix culture of iron-oxidating bacteria (most prevalent genus *Acidithiobacillus*) | static | (electric field) 40 mA, 0.2–2.7 V | - | 2–6 days | [69] |
| bacterial inactivation | promotion | *Salmonella enteritidis* | pulse | (electric field) 5–50 kV cm$^{-1}$, 2 kW | | 50 ns to 3 μs | [70] |
| quality of frozen food | promotion | various food products | combination of static and pulsed | 480 | 20 KHz | 1443 ± 2 s | [71] |
| aging of Baijiu | promotion | Feng-flavored Baijiu (fermentation product) | static | 210 | - | 24 h | [72] |
| bacterial destruction | promotion | *Staphylococcus aureus* | alternating | not stated | 15 kHz | 20 min | [73] |
| antibiotics resistance | reduction | *Staphylococcus aureus* | alternating (rotatory) | 8.1 | 5–50 Hz | 120 h | [74] |
| biofilm formation | reduction | *Pseudomonas aeruginosa* | alternating (rotatory) | 23–34 | 10–50 Hz | 5 min | [75] |
| | reduction | *Staphylococcus aureus* | static + magnetics nanoparticles | ≈1000 | - | 1 min | [76] |
| metal bioremediation (Al, Cu, Pb) | promotion | mixed culture (industrial sludge) | alternating | not stated | 5 Hz | 6–12 days | [77] |

* COD (chemikal oxygen demand), ** organochlorine pesticides, polycyclic aromatic hydrocarbons (PAH), heavy metals and organotins.

## 5.1. Wastewater Treatment

According to multiple authors, the static magnetic field is a promising innovative method for improving the biodegradation of organic substances and nitrogen removal [54,55,78]. Hou et al. [55] examined its influence on the wastewater treatment process under both aerobic and anaerobic conditions. The impact was measured by the

presence of oxidized/reduced coenzymes and the total nitrogen consumption. In the wastewater treatment reactor, after exposing the biomass to a static magnetic field (8 and 12 h, 39.5–65.3 mT), a significant increase in nitrogen removal (>80%) was observed compared to that in the control. Another effect was the increase in the NADH/NAD$^+$ ratio and, consequently, the activity of the electron transport system. The contents of NADH and the electron transport system affect not only the nitrogen removal efficiency but also the accumulation of denitrification intermediates [54]. Similar results were reported by Hou et al. [55] when testing a static magnetic field (30 mT) induced by a magnet. This magnetic field enhanced the denitrification efficiency, particularly in wastewater with low C/N ratios, resulting in an average increase in NO$_3^-$ removal of 6.58%. The static magnetic field also had a significant impact on the stability of the microbial community structure, favoring the denitrification of the bacteria stability.

Results from Zaidi et al. [53] indicate that the efficiency of nitrification is consistently higher during the application of a static magnetic field (88 mT). Specifically, ammonia removal increased by 90% and nitrite removal increased by 74–81% compared to the control. The authors explained these observations by the improved properties of activated sludge biomass due to the induced magnetic field. The metabolic activity of aerobic bacteria was also increased compared to the control, as evidenced by the average oxygen uptake rate ranging from $11.7 \pm 1.2$ mg/L/h compared to the control ($9.5 \pm 0.4$ mg/L/h). According to these authors, the static magnetic field provides control over the biomass activity [53].

The influence of magnetic fields on improving wastewater treatment processes has also been observed in the case of photosynthetic bacteria [56]. At 0.35 T, the lowest value of COD (Chemical Oxygen Demand) was recorded after 48 h, one day earlier than in the control groups. However, unlike previous studies, Lu et al. [56] did not demonstrate an effect on the removal of NH$_4^+$ ions.

Regarding optimal exposure duration testing, the most suitable period appears to be between 0.5 days and 3–4 days, which enhances the settling ability of activated sludge. Nevertheless, longer exposure to the magnetic field (>6 days) leads to the accumulation of dead or inactive microorganisms in the system, thus affecting the sludge activity. This negative state promotes the growth of filamentous microorganisms (*Nocardioforms*, *Gordonia* sp., and *Microthrix parvicella*), resulting in a high risk of bulking sludge occurrence [79].

*5.2. Pollutant Removal*

One of the main goals of bioremediation is to use environmentally friendly procedures with high contaminant removal rates. Sites contaminated with multiple categories of contaminants can be challenging candidates for bioremediation. In such cases, magnetic fields offer the opportunity to activate the metabolism of bacteria naturally present in the environment, resulting in an accelerated bioremediation process [80].

After exposure to a magnetic field (200 mT), *Pseudomonas stutzeri*, *Cupriavidus metalliduras*, and *Rhodococcus equi*, which are naturally found in contaminated environments, exhibited the ability to degrade over 90% of the selected contaminants [57].

Qu et al. [58] demonstrated a potential improvement in anaerobic lignocellulose degradation in manure with simultaneous biogas production using an electromagnetic field (230–260 mT, electric field with a voltage of 0.3–0.8 V). The degradation rate of cellulose was increased by 125%, and the degradation rate of lignin was increased by 203%. According to the authors, this anaerobic bioremediation technology not only addresses environmental pollution issues but also produces biogas and high-quality fertilizer.

Another area that has received limited exploration is the possibility of enhancing the biological removal of oil in the event of its release into the environment. Ren et al. [59] investigated the effect of a static magnetic field on oil removal by the *Acinetobacter* sp. strain. The results showed that low-intensity magnetic fields (15–35 mT) improved the ability of *Acinetobacter* sp. B11 bacteria to remove oil by 11.9% at 25 mT compared to the control. This magnetic field density provided a sufficient increase in the membrane permeability

without damaging it and enhanced the activity of superoxide dismutase (SOD), effectively boosting the bacteria's ability to degrade oil.

### 5.3. Food Industry Applications

Magnetic fields, due to their presumed positive effects, have the potential for wide application in the food industry, as they do not alter the resulting chemical composition of the product [60]. However, exposure to a magnetic field can interfere with reaction kinetics by increasing the collision rate between chemical substrates or enhancing diffusion rates [81].

Weaker magnetic fields (in range of mT) generally support microbial growth [82–84], while high-intensity pulsed magnetic fields are widely used as non-thermal sterilization technologies in food processing [70,85,86]. These technologies are associated with short sterilization times and low energy consumption and are better at preserving the original nutritional and sensory qualities of food. Other areas where magnetic fields are already being utilized include enzyme activity passivation [60], the preservation of fruits and vegetables [71,87], and the improvement of the fermentation process [88].

Most fermented foods require a natural aging process to develop the desired flavors and aromas, which is often a barrier to cost-effective production. Therefore, a process that enhances the desired taste in a shorter period of time would be highly beneficial. Previous studies have explored the use of magnetic fields for assisting microbial fermentation [88].

Recently, favorable results have been observed with the application of various magnetic field strengths to accelerate the aging of a wide range of fermented foods [72]. The impacts of variable magnetic fields (0–5 mT) on the fermentation process and vinegar aging have also been described [60]. The authors used ultrasound and alternating magnetic fields to accelerate the vinegar aging process. The most significant acceleration effect was observed when ultrasound and magnetic fields were combined, but even the use of magnetic fields alone had an accelerating effect on the aging process. The experimental sample exhibited the presence of approximately 33 volatile compounds, including alcohols, acids, aldehydes, ketones, esters, heterocycles, and others. The content of furfuryl formate observed after a 3 h exposure to the magnetic field typically occurs naturally in aged vinegar after 60 to 120 months, confirming the authors' hypothesis about the positive effects of the magnetic field on vinegar aging.

Another potential application of magnetic fields can be found in the production of dietary supplements. According to Gu et al. [67], *Spirulina* cells exposed to a magnetic field (60 mT) exhibited an unchanged protein content, but the concentration of carbohydrates decreased by 69.1%. This indicates that the biomass obtained through this method has the potential to be used in the development of protein supplements with a low sugar content, making them suitable for diabetic patients.

### 5.4. Medical Applications

Magnetic fields have potential applications in medicine as a means of sterilization (high-frequency fields) [73], as well as being a mechanism to increase the sensitivity of pathogens to antibiotics [52,74] and remove bacterial biofilms from surgical instruments [61].

The application of rotating magnetic fields enhances the susceptibility of the bacteria biofilms of *S. aureus* and *P. aeruginosa* to antimicrobial agents (such as gentamicin, ciprofloxacin, octenidine, chlorhexidine, polyhexanide, and ethacridine lactate) [75].

Magnetic fields are often used in conjunction with magnetic nanoparticles, for example, for the removal of biofilms. It has been documented that magnetic nanoparticles penetrate deeply into the biofilm and create artificial channels within its matrix, resulting in 4- to 6-fold faster biofilm removal by facilitating the enhanced penetration of antibiotics through these artificial channels [76]. Antibiotics, in combination with magnetic fields and silver nanoparticles, were also used in the study by Wang et al. [89], who observed that when a silver nanoparticle was coated with a layer of the antibiotic gentamicin and guided into the biofilm using a magnetic field (200 mT), gradual release of the antibiotic occurred during

penetration into the biofilm, thereby increasing the effectiveness of biofilm removal. The inhibition of biofilm growth was also observed depending on the pH. Fan et al. [62] found that exposing the biofilm to a static magnetic field (170 mT) for 24 to 72 h while adjusting the pH to 9 resulted in the most effective biofilm removal.

Brkovic et al. [63] observed the effect of a magnetic field on the reduction of microorganisms in dental plaque in vitro. A significant reduction was observed during the exposure to the magnetic field over a 24 h period. Magnetic fields induced by permanent micro-magnets can be used as adjunctive therapies for the treatment of periodontal tissue diseases in the oral cavity.

### 5.5. Metabolism Regulation and the Activation of Enzymes

Armenia et al. [90] investigated selected bacterial enzymes ($\alpha$-amylase and L-aspartate oxidase) conjugated with iron oxide nanoparticles. These nanoparticles have the ability to increase their temperature in the presence of a strong magnetic field (25.2 mT, frequency 829 kHz), enabling the remote control of enzyme activity. Thermophilic enzymes chemically bound to nanoparticles were remotely activated by a locally increased temperature for 30 min, without heating the entire medium. The nanoactivation of thermophilic enzymes using a magnetic field has potential applications in various areas. Multienzyme processes involving enzymes with different temperature optima could be carried out in the same reaction vessel. Additionally, thermolabile products could be efficiently produced using thermophilic enzymes without compromising their stability.

Furthermore, an increase in enzyme activity was observed with an increasing frequency of the applied variable magnetic field. A higher applied field frequency resulted in a greater heat increase generated by the nanoparticles, leading to greater activation of the immobilized enzyme. These findings demonstrate the possibility of conducting biotechnological multienzymatic processes in which one of the enzymes/substrates/products is thermolabile, while the other requires a high temperature for its activity.

The activation of enzymes can accelerate specific processes through which microorganisms produce valuable products with potential industrial applications. For example, microbial fermentation plays a crucial role in hydrogen production. When a magnetic field (3.2 mT) was applied to the glucose fermentation system by *Clostridium pasteurianum*, an increase in hydrogen production of 366% compared to the control was observed. Studying how magnetic fields interact with microorganisms can also improve hydrogen production, which is becoming increasingly valuable in modern industries [64].

### 5.6. Metal Recovery

Bioleaching is an alternative method for extracting metals from electronic waste or low-grade ores containing metals such as zinc, nickel, lead, cadmium, copper, or cobalt [91,92]. Typical bacteria used in bioleaching include those from the genera *Acidithiobacillus*, *Thiobacillus*, *Thermithiobacillus*, *Leptospirillum*, *Halothiobacillus*, and *Sulfolobus* [93,94] but *Acidithiobacillus ferrooxidans* remains the most-studied one. *A. ferrooxidans* is a chemolithoautotrophic Gram-negative $\gamma$-proteobacterium that fixes $CO_2$ from the atmosphere and obtains energy through the oxidation of iron ($Fe^{II}$), elemental sulfur or partially oxidized sulfur compounds [95]. It can be cultivated on inexpensive and uncomplicated media [56]. *A. ferrooxidans* has potential applications in the desulfurization of solids and gases, metal recovery from ores, e-waste, and sludge [95], as well as the simultaneous production of certain products with industrial potential [96].

Iron- and sulfur-oxidizing bacteria are known for their slow growth rates. The slow reaction rate and long operation time of the bioleaching process somewhat limit its practical application [69], which presents the potential for biotechnological improvements, such as through the use of a magnetic field. Currently, electromagnetic fields are used for the magnetic separation of substrates or bioleaching products [97], or the substrate for bioleaching itself can be a material with magnetic properties [98].

In the past, it has been demonstrated that even some non-magnetotactic bacteria, including *Magnetospirillum aberrantis*, *A. ferrooxidans*, and *Leptospirillum ferriphilum*, can perceive a magnetic field and produce magnetosomes [65,99]. It is also known that *A. ferrooxidans* does not exhibit magnetotaxis in the presence of the Earth's magnetic field, while weak magnetotaxis can be observed in an artificially induced magnetic field [65].

Only a small number of studies have focused on the use of a magnetic field to increase the efficiency of the bioleaching process. Static magnetic fields (3.15 mT) were tested, and the ability to increase the copper bioleaching yield from pyrite by 15.8% to 18.9% was observed during a 25-day cultivation period [67]. Similar results were achieved using static magnetic fields (2 mT, 5 mT, 8 mT, and 11 mT) during the bioleaching of arsenic and cadmium, where the yield was higher by 2% to 8% compared to the control [66].

Le et al. [68], who also observed the stimulation of copper bioleaching using a static magnetic field (9.6 mT), pointed out other important parameters that need to be optimized for the desired outcome. These parameters include adjusting the pH to 2, using an inoculum concentration of 11.3%, and using a bioleaching temperature of 27.1 °C, which is lower than what other authors have reported [65,66]. This difference could be attributed to the use of a mixed culture of *Acidithiobacillus ferrooxidans* and *Acidithiobacillus thiooxidans* by Le et al. [68], in contrast to other authors who used a pure culture of *A. ferrooxidans* [66,77].

According to the mentioned authors, static electromagnetic fields can alter the microstructural properties of $H_2O$ molecules and the microbial physiological characteristics of reactors, as well as increasing the permeability of the cytoplasmic membrane [68]. Another explanation could be related to the increased solubility of oxygen and salts in water after exposure to a magnetic field, which could contribute to improved conditions for aerobic cultivation. Additionally, magnetic fields can affect microorganisms by alternating the intracellular signaling pathways of various charged macromolecules and ions in cells specifically for different microbial taxa [58].

In addition to the magnetic field, the effect of an electric current on the enhancement of the bioleaching process was also investigated. An electric field (40 mA, 0.8 V to 2.7 V) induced by a cathode and anode directly into the growth medium accelerated the bioleaching process of copper from electronic waste and the simultaneous metal recovery from the leachate. The electric field itself did not have an impact on the leaching process without the use of bacteria, but with the addition of a mixed bacterial culture, complete copper leaching was achieved within 3 days, which is 2 days faster than bioleaching without the electric field. Increased metabolic activity of the bacteria and a 60% to 106% increase in the ATP concentration compared to the control were also observed [69].

## 6. Conclusions

The usage of living organisms in industrial applications offers environmentally acceptable alternatives to traditional methods and chemical synthesis processes. However, the control of biotechnological processes poses significant challenges for today's industries. The complexity of biological systems, the variability in organisms and conditions, non-linear dynamics, the lack of knowledge, up-scaling difficulties, cost considerations, time constraints, ethical and safety concerns, and the extensive industrial applications of biotechnology all contribute to the complexity of process control. To address these challenges, the concept of control in biotechnological processes encompasses ensuring the intended procedure is followed, resolving issues that arise, and ensuring safety and environmental protection. Current control methods involve the use of various substrates, the monitoring of external conditions, process automation, the use of specific microorganism strains, and genetic engineering.

This review demonstrates that magnetic fields exert distinct effects on prokaryotic and eukaryotic organisms. A shared impact has been observed in terms of accelerating cellular metabolism, with potential applications in pollution remediation and the food industry. However, disparities arise not only in cellular metabolism but also in the overall influence of magnetic fields on multicellular organisms composed of millions of cells. In this context,

magnetic fields may influence the transport systems in plants, offering potential utility in phytoremediation. Prokaryotic organisms exhibit heightened responsiveness to relatively weaker magnetic fields due to their simple cell structures, rapid reproduction rates, and genetic adaptability. This explains the prevalent focus of research papers on prokaryotic organisms, owing to their superior adaptive capacity to environmental stressors such as magnetic fields.

Emerging evidence suggests that electromagnetic fields could be a tool for influencing biological processes. The sustainability of magnetic field applications in biotechnological processes represents a promising avenue for enhancing the control and efficiency of these complex processes. Current methods for managing biotechnological processes are limited to substrate variations, external condition monitoring, process automation, specific microbial strains, and genetic engineering. However, the integration of living organisms into industrial applications has emerged as an environmentally friendly and sustainable alternative to conventional chemical syntheses. Magnetic fields show potential to accelerate growth and biomass production, stabilize bacterial communities, and expedite processes like waste processing and metal recovery. However, there is limited practical usage of this potential, and further research is needed. Implementing effective control mechanisms is crucial to make biotechnological processes competitive alternatives to traditional industrial procedures. Exploring the potential of magnetic fields to control biotechnological processes offers a promising avenue for future investigation. By using magnets or weak electric fields, magnetic fields can contribute to long-term sustainability while enhancing the competitiveness of biotechnological processes compared to less ecologically friendly methods. This holds great promise for the advancement of sustainability in biotechnological applications.

Furthermore, magnetic fields have shown potential for use in medical and laboratory applications. In cancer treatment, circularly polarized magnetic fields have been explored to enhance the immune response, promote tumor cell death, and facilitate the maturation of dendritic cells and infiltration of T lymphocytes. Additionally, magnetic fields have been found to have beneficial effects on wound healing, arthritis treatment, and stem cell differentiation and migration and can be also used in environmental biotechnologies in processes of bioleaching and metal waste removal.

In conclusion, the use of electromagnetic fields shows promise for improving various industrial applications involving prokaryotic organisms. Compared to eukaryotic organisms, prokaryotic organisms offer advantages such as simpler cellular structures, rapid responses to environmental changes, and wide industrial applications, which has led to increased research.

**Author Contributions:** M.S. wrote the paper. J.S.-K., A.L., P.J. and I.M. edited the manuscript. All authors have read and agreed to the published version of the manuscript.

**Funding:** The work was supported by financial by projects VEGA 1/0018/22 and 2/0108/23, APVV-21-0504 and FPPV-32-2023.

**Data Availability Statement:** The manuscript has no associated data.

**Conflicts of Interest:** The authors declare no conflict of interest. The funders had no role in the design of the study; in the collection, analyses, or interpretation of data; in the writing of the manuscript; or in the decision to publish the results.

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
