# Peer review of "Application of a Magnetic Field to Enhance the Environmental Sustainability and Efficiency of Microbial and Plant Biotechnological Processes"

_sustainability, doi:10.3390/su151914459_

Round 1

Reviewer 1 Report

General comment:

Thank you for giving me an opportunity to review the manuscript “Application of magnetic field to enhance environmental sustainability and efficiency of microbial and plant biotechnological processes”. In this review article, the authors explained that electromagnetic fields offer a great opportunity to control various biotechnological processes by altering growth and cell metabolism. In general, the topic taken by authors is novel and timely and provides a comprehensive overview of applications of magnetic fields in microbial and plant biotechnological processes. After a careful reading, I found that this manuscript can be accepted after minor revisions. My specific comments are given below:

Specific comments:

1.      In the abstract section, please elaborate on the criteria and sources of the literature review. How many papers from which databases were reviewed? A specific protocol must be provided in both the abstract and methodological parts.

2.      What were the criteria for the selection of particular studies in this review? What was the timeframe of study selection (from to year)?

3.      At the start of the introduction section, the authors stated that Earth has a magnetic field of Bg≈ 50 μT, however, please specify that this value differs greatly at the equator and poles.

4.      Please briefly describe the role of the outer shield of the magnetic field in controlling harmful solar and cosmic rays.

5.      The statement in line 39 requires reference to the name of organisms. Please be specific here.

6.      Lines 77-85: no reference?

7.      Authors could convert the points under the control of biotechnological processes as a figure and outline the state of the art of magnetic field in this context.

8.      Line 268-270: reference required.

9.      Two headings: pollutant removal and contaminant removal must be corrected. Please be consistent with any one format.

10.   Overall, the English of the manuscript should be checked by a native writer.

Overall, the English of the manuscript should be checked by a native writer.

Author Response

  1. In the abstract section, please elaborate on the criteria and sources of the literature review. How many papers from which databases were reviewed? A specific protocol must be provided in both the abstract and methodological parts.

Thank you for your kind reminder, the Abstract and Metods have been edited as suggsted.

  1. What were the criteria for the selection of particular studies in this review? What was the timeframe of study selection (from to year)?

Thank you for your kind reminder, Metods section have been edited as suggsted.

  1. At the start of the introduction section, the authors stated that Earth has a magnetic field of Bg≈ 50 μT, however, please specify that this value differs greatly at the equator and poles.

Thank you for your coment, , text have been modified as requested

  1. Please briefly describe the role of the outer shield of the magnetic field in controlling harmful solar and cosmic rays. The statement in line 39 requires reference to the name of organisms. Please be specific here.

Thank you for your coment, the introduction section has been modified as requested

  1. Lines 77-85: no reference?

Thank you for your kind reminder, the references must have get lost during final editing. The mistake have been corected.

  1. Authors could convert the points under the control of biotechnological processes as a figure and outline the state of the art of magnetic field in this context.

The descriptive picture depicting the challenges of biotechnology process control has been added to the second chapter.

  1. Line 268-270: reference required.

Thank you for your kind reminder, the references must have get lost during final editing. The mistake have been corected.

  1. Two headings: pollutant removal and contaminant removal must be corrected. Please be consistent with any one format.

The format have been unified.

  1. Overall, the English of the manuscript should be checked by a native writer.

Thank you for your kind reminder, the english was additionaly checked by our collegues with more experimences in this field.

Reviewer 2 Report

Through the collection and analysis of literature on the influence of magnetic field on microorganisms, this study shows that electromagnetic field may be a tool to affect biological processes, and the data are sufficient and reliable, providing certain basis for future research on the potential of magnetic field in controlling biological processes.The specific modification opinions are as follows:

1.  The chapter number is incorrect.

2.  Try not to segment the abstract, it is recommended to modify.

3.  Introduction:The usage of magnetic fields can also present a long-term sustainable  alternative that is environmentally acceptable (while using magnets or weak electric fields) and has the potential to influence a broad spectrum of biotechnological processes,thereby enhancing their competitiveness compared to traditional less ecological methods.Lack of literature support.

4.  The alcohol fermentation process spontaneously ends when the ethanol concen-tration in the medium reaches approximately to 20%. Such high levels of ethanol have an inhibitory effect on the yeast cells that produce it. The inhibitory effect of ethanol lies in changes of the organization and permeability of the cell membrane and deactivation of cytosolic enzymes [1].Reference insertion error.

5.  It is suggested to sort out the summary part, point description, and supplement the similarities and differences between eukaryotes and prokaryotes.

6.  It is suggested to add the specific information of the influence in Table 1 and Table 2, so as to more intuitively show the degree of influence of the magnetic field on microorganisms.

7.  The topic of this paper is about the sustainability of the technical process of magnetic field on plant cells and animal cells, and it is suggested to further summarize and explain in the summary part.

8.  In this paper, the research on the effect of magnetic field on the technical process of plant cells is insufficient, and it is suggested to supplement.

9.  Too few charts, suggested to add.

Author Response

  1. The chapter number is incorrect.

 Thank you for your kind reminder, the mistake have been corected.

  1. Try not to segment the abstract, it is recommended to modify.

Thank you for your coment, the iAbstract section has been modified as requested

  1. Introduction:”The usage of magnetic fields can also present a long-term sustainable alternative that is environmentally acceptable (while using magnets or weak electric fields) and has the potential to influence a broad spectrum of biotechnological processes,thereby enhancing their competitiveness compared to traditional less ecological methods.”Lack of literature support.

 Thank you for your kind reminder, the references must have get lost during final editing. The mistake have been corected.

  1. “The alcohol fermentation process spontaneously ends when the ethanol concen-tration in the medium reaches approximately to 20%. Such high levels of ethanol have an inhibitory effect on the yeast cells that produce it. The inhibitory effect of ethanol lies in changes of the organization and permeability of the cell membrane and deactivation of cytosolic enzymes [1].”Reference insertion error.

Thank you for your kind reminder, the references must have get lost during final editing. The mistake have been corected.

  1. It is suggested to sort out the summary part, point description, and supplement the similarities and differences between eukaryotes and prokaryotes.

Thank you for yours suggestion, the summar part have been added to the Conclusion section.

  1. It is suggested to add the specific information of the influence in Table 1 and Table 2, so as to more intuitively show the degree of influence of the magnetic field on microorganisms.

We appreciate your kind reminder, but in our view, the tables are already quite extensive, and adding more specific information might actually reduce clarity. We've made sure to include all the necessary additional details in the text that follows immediately after the mentioned tables.

  1. The topic of this paper is about the sustainability of the technical process of magnetic field on plant cells and animal cells, and it is suggested to further summarize and explain in the summary part.

Thank you for yours suggestion, the additional text about sustainbility of the megnetic field have been added to the Conclusion section.

  1. In this paper, the research on the effect of magnetic field on the technical process of plant cells is insufficient, and it is suggested to supplement.

 Thank you for your suggestion. The research field related to plant biotechnology is still relatively unexplored, which is evident in the limited number of published papers. This is reflected in the length of the plant section within the manuscript.

  1. Too few charts, suggested to add.

Thank you for your suggestion, we added descriptive picture depicting the challenges of biotechnology process control as part of  second chapter.

Reviewer 3 Report

This paper is a review-type manuscript which retrospect the application of magnetic field to enhance environmental sustainability and efficiency of microbial and plant biotechnological processes, from the following two main aspects, one is using eukaryotic organisms and another one is involving prokaryotic organisms. Totally speaking, this topic is interesting and the paper is well-organized. Therefore, this reviewer suggests the acceptance after revisions according to the following comments. 

1. In the Introduction, the aim why the authors write such a review is not very clear. It is suggested the author further elaborate on why such a review is carried out.

2. Conclusion section is insufficient. The main problems in the existing researches and the future research direction need to be further condensed and summarized. A good idea is to adopt the mode of point by point, thus interested readers can see the future research direction and possible hot issues at a glance from the Conclusion.

3. In addition, Introduction and Conclusion should be numbered as a Section title.

4. Section 2 is not found in this paper, Section 1 is followed by Section 3, please check it.

5. The text at lines 204-210 is the same as the text at lines 223-229, please check it.

6. As a review paper, the number of References is large, so the author must organize them carefully and make them fit the style of the journal. For example, the all authors’ names should be listed, and journal names should be the abbreviated forms.

English is OK.

Author Response

  1. In the Introduction, the aim why the authors write such a review is not very clear. It is suggested the author further elaborate on why such a review is carried out.

Thank you for your suggestion. This text was also added as a part of the Introduction section.

  1. Conclusion section is insufficient. The main problems in the existing researches and the future research direction need to be further condensed and summarized. A good idea is to adopt the mode of point by point, thus interested readers can see the future research direction and possible hot issues at a glance from the Conclusion.

Thank you for yours suggestion, the additional text about sustainbility and summarising part about the effect of magnetic field on on prokaryotic and eucaryotic organismsm have been added to the Conclusion section.

  1. In addition, Introduction and Conclusion should be numbered as a Section title.

Thank you for your suggestion, the chapter number for Introduction and Conclusion section have been added.

  1. Section 2 is not found in this paper, Section 1 is followed by Section 3, please check it.

The chapters numbers have been corrected, thank you for your reminder.

  1. The text at lines 204-210 is the same as the text at lines 223-229, please check it.

Thank you for your kind reminder, the error was corrected.

Round 2

Reviewer 3 Report

In the Report 1, there is still one comment that the authors have not addressed, so minor revisions are necessary.

6. As a review paper, the number of References is large, so the author must organize them carefully and make them fit the style of the journal. For example, the all authors’ names should be listed, and journal names should be the abbreviated forms.

English is OK.

Author Response

  1. As a review paper, the number of References is large, so the author must organize them carefully and make them fit the style of the journal. For example, the all authors’ names should be listed, and journal names should be the abbreviated forms.

I apologize for not addressing your comment in the first round of revisions. In the newly uploaded manuscript, you will find that all the references have been formatted as per the journal's requirements. Thank you for your feedback.